# Transcranial Magnetic Stimulation for the Treatment of Cocaine Addiction: A Systematic Review

**DOI:** 10.3390/jcm10235595

**Published:** 2021-11-28

**Authors:** Alezandra Torres-Castaño, Amado Rivero-Santana, Lilisbeth Perestelo-Pérez, Andrea Duarte-Díaz, Ana Toledo-Chávarri, Vanesa Ramos-García, Yolanda Álvarez-Pérez, Javier Cudeiro-Mazaira, Iván Padrón-González, Pedro Serrano-Pérez

**Affiliations:** 1Canary Islands Health Research Institute Foundation (FIISC), 38109 El Rosario, Spain; amado.riverosantana@sescs.es (A.R.-S.); andrea.duartediaz@sescs.es (A.D.-D.); anatoledochavarri@sescs.es (A.T.-C.); vanesa.ramosgarcia@sescs.es (V.R.-G.); yolanda.alvarezperez@sescs.es (Y.Á.-P.); 2Evaluation Unit of the Canary Islands Health Service (SESCS), 38019 El Rosario, Spain; lilisbeth.peresteloperez@sescs.es; 3The Spanish Network of Agencies for Health Technology Assessment and Services of the National Health System (RedETS), 28071 Madrid, Spain; 4Galician Brain Stimulation Center, 15009 A Coruña, Spain; javier.cudeiro@udc.es; 5Neuroscience and Motor Control Group (NEUROcom), Instituto Biomédico de A Coruña (INIBIC), Universidad de A Coruña, 15006 Oza, Spain; 6Institute of Neuroscience, University of La Laguna, Guajara Campus, 38200 San Cristobal de La Laguna, Spain; ivpadron@ull.edu.es; 7Group of Psychiatry, Mental Health and Addictions at the Vall d’Hebron Institut de Recerca (VHIR), 08035 Barcelona, Spain; pedrogserrano@gmail.com

**Keywords:** cocaine use disorder, craving, non-invasive brain stimulation, transcranial magnetic stimulation, systematic review

## Abstract

Long-term cocaine use is associated with cognitive deficits and neuro-psychiatric pathologies. Repetitive transcranial magnetic stimulation (rTMS) is an emerging therapeutic strategy relating to changes in brain activity. It stimulates the prefrontal cortex and is involved in inhibitory cognitive control, decision making and care. This systematic review aims to evaluate and synthesize the evidence on the safety, effectiveness, and cost-effectiveness of rTMS for the treatment of cocaine addiction. A systematic review of the literature was carried out. The following electronic databases were consulted from inception to October 2020: MEDLINE, Embase, CINAHL, PsycINFO, Cochrane Central Register of Controlled Trials and Web of Science. Randomised controlled trials, non-randomised controlled trials and case-series and full economic evaluations were included. Twelve studies were included. No identified study reported data on cost-effectiveness. Significant results of the efficacy of TMS have been observed in terms of the reduction of craving to consume and the number of doses consumed. No serious adverse effects have been observed. Despite the low quality of the studies, the first results were observed in terms of reduction of cocaine use and craving. In any case, this effect is considered moderate. Studies with larger sample sizes and longer follow-ups are required.

## 1. Introduction

Cocaine use disorder (CUD) is a significant health problem, with about 12–21 million users worldwide in 2014 [1,2]. Chronic cocaine use can cause damage and changes to the prefrontal cortex (PFC) [3], including a significant reduction in brain volume [4,5], cortical hypoactivity [6,7], impaired executive functions, and dysregulation of neurotransmitter systems [8,9,10]. Preclinical studies have shown that loss of inhibitory control, resulting from damage to the PFC, appears to be crucial in compulsive drug-seeking behaviours [11,12] and intense and uncontrollable craving from consuming a substance [13]. This desire is one of the key characteristics of substance dependence, which has been shown to be one of the most important contributors to relapse. Several types of evidence indicate that substance dependence involves the dopaminergic system, causing a hypodopaminergic state in the mesolimbic system [14].

Previous research has described the neural network distributed in the two hemispheres present in the pathophysiology of craving, involving the nucleus accumbens, the amygdala, the anterior cingulum, the orbitofrontal cortex and the dorsolateral prefrontal (DLPFC) [15,16]. The DLPFC, specifically, participates in the reward, motivation and decision-making circuits that provide the substrate for the integration of cognitively and motivationally relevant information and the inhibitory control over the options of an immediate reward [17]. However, the poor functioning of the DLPFC and the anterior cingulate cortex may explain a reduction in inhibitory control of behaviour and a tendency to relapse into the use of alcohol and other drugs [18]. In fact, the most recent technological advances, using optogenetic techniques that allow the manipulation of neuronal groups in a very effective and localized way, have made it possible to delineate the cortico-subcortical circuits that are related to addictions in animals [12,19]. In these experiments, a hypofunction of the prefrontal cortex was related to a lack of subcortical inhibitory control (over the reward circuits, see Section 4).

Following this hypothesis, Terraneo et al. [13] designed an experiment in which the activation of the left dorsolateral prefrontal cortex, by means of rTMS, could, in patients addicted to cocaine, reduce consumption and craving.

To date, an effective treatment for cocaine addiction has not been found [20], and currently there are new treatments in experimental research [21]. Neuromodulation techniques, such as transcranial magnetic stimulation (TMS), have been investigated as potential treatments with fewer side-effects and contraindications than drugs for substance use disorders, and is therefore a promising therapeutic alternative to conventional pharmacotherapy and behaviour modification therapies [22]. 

TMS is a non-invasive human brain neuromodulation technology based on the principle of electromagnetic induction. The transient application of an electric current to a coil of conductive material produces a time-varying magnetic field, capable of inducing an electric field at a distance, affecting the electrical activity of neurons in the cerebral cortex [23,24]. This electric field must be of sufficient magnitude for neuronal depolarization to occur, followed by an increase in synaptic activity and the production of action potentials [25]. The extent of the induced field depends on the geometry and size of the coil used in the TMS equipment. 

Several coils with different shapes have been designed to stimulate different regions of the brain, the most common being circular coils, which allow large areas of the cortex to be affected, and those with a figure of eight allowing more focal stimulation. Both coils are useful for superficial cortical stimulation (about 2–3 cm deep) [26]. However, in certain situations it may be useful to reach deeper targets, for which special coils have been developed, such as double cone coils that can reach a depth of 3–4 cm and the so-called H-type coils, which can reach depths up to 6 cm [27]. This is what is known as deep brain stimulation TMS (dTMS). Currently, several manufacturers provide coils with specific characteristics to achieve a focused or deep stimulation [28]. 

Repetitive TMS (rTMS) at low frequency (≤1 Hz) has inhibitory effects [29], whereas high frequency rTMS (>5 Hz) is excitatory [13,30]. The rTMS uses a coil held against the scalp and located over the cortical zone of choice. Stimulation is performed using repetitive stimulation in the form of continuous pulses at a certain frequency, or repetitive trains of magnetic pulses which affect a specific area of the brain and those other areas that are connected [22]. Stimulation parameters of rTMS can vary significantly with respect to stimulus intensity, total number of pulses, and frequency. These variations aim to customize rTMS parameters and improve inhibitory processes, which can be abnormal in substance abuse cases (i.e., lack of impulse control or impulsivity). On the other hand, rTMS has been shown to be highly effective in studying the excitation-inhibition balance in a specific area of the brain. In order to do this, different cortical inhibition and excitation protocols have been developed with single-pulse TMS or paired pulses [31,32]. 

Different temporal patterns of stimulation have been developed. This is the case of Theta Burst Stimulation (TBS) [33], which involves bursts of three pulses at 50 Hz that are repeated at theta frequency (5/s). There are two types of TBS with opposite effects: intermittent TBS (iTBS) has an excitatory effect that lasts over time. This pattern is repeated for 190 s, implying that the subject is supplied with 600 pulses in total [33,34]. The other protocol is known as continuous TBS (cTBS), which induces inhibitory effects that also last over time, involving a transient depression of neuronal behaviour long-term. In this case the stimulation involves a 40-s train of uninterrupted TBS (600 pulses) [15,33]. rTMS in any of its forms is a painless procedure and its common side effects, if they occur, are generally minor, although seizures have been reported very occasionally; thus, most patients tolerate it very well [35,36]. Consequently, rTMS has been suggested as a possible alternative treatment for substance use disorders, such as cocaine, which is currently being investigated [22].

This systematic review (SR) aims to evaluate and synthesize the evidence on the safety, effectiveness, and cost-effectiveness of TMS for the treatment of cocaine addiction.

## 2. Materials and Methods

An SR of the literature was carried out in accordance with the Preferred Reporting Items for Systematic reviews and Meta-Analysis (PRISMA) Statement [37]. The detail of the PRISMA checklist can be found in Appendix A. This SR was registered in the International Prospective Register of Systematic Reviews (PROSPERO) with the number CRD42021233283.

### 2.1. Search Strategy

The following electronic databases were consulted, from inception to October 21st 2020: MEDLINE, Embase, CINAHL, PsycINFO, Cochrane Central Register of Controlled Trials and Web of Science (WOS). In addition, a manual consultation of references in non-indexed health journals and other relevant health websites was performed. Search terms were grouped around the following terms: “cocaine use disorder, substance abuse disorders, craving, transcranial magnetic stimulation”. As an example, the MEDLINE search strategy is shown in Table 1. Search strategies for the other five electronic databases are available in Appendix A. No language or publication year restrictions were applied to limit the search.

Additionally, manual searches were carried out on clinical trail.com to identify ongoing studies.

### 2.2. Inclusion and Exclusion Criteria

#### 2.2.1. Design

Randomised controlled trials (RCTs), non-randomised controlled trials (nRCTs), case-series and full economic evaluations (EE) published in English or Spanish were included. Qualitative studies, conference abstracts, letters, commentaries, essays, and book chapters were excluded.

#### 2.2.2. Population

Studies addressing subjects with cocaine dependence or CUD, seeking treatment or not, were included. Thus, studies with participants taking opioids, cannabis, tobacco, alcohol, and food abusers or non-abusers were excluded. Studies with mixed use were included as long as the effect on cocaine use and craving was evaluated, and the results were reported separately.

#### 2.2.3. Intervention

Studies applying any rTMS protocol were included. Studies with a single session were excluded, unless they used more recent stimulation protocols such as deep rTMS or TBS.

#### 2.2.4. Comparator

The main comparators considered were sham-stimulation, waiting list, pharmacological treatment, no treatment and treatment as usual.

#### 2.2.5. Outcomes

The primary effectiveness outcomes were the reduction of cocaine use or relapses, evaluated by laboratory analyses (e.g., urine, hair), self- and hetero report. Secondary outcomes included reduction of craving, addiction severity, anxiety, depression, and sleep quality. Regarding, safety, the main outcomes were serious and non-serious adverse events. The incremental cost-effectiveness ratio (ICER) was the cost-effectiveness outcome.

### 2.3. Study Selection

Bibliographic references were stored using the Reference Manager Edition Version 10^©^ (Thomson Scientific, EE.UU.). Electronic search results were downloaded into a standardized Excel datasheet and duplicates were removed. Titles and abstracts were screened first. Subsequently, those articles selected as relevant were full text reviewed to determine whether a study met the inclusion criteria. This screening process was conducted independently by two reviewers. Any doubt or disagreement was resolved by discussion and, when necessary, with the participation of a third reviewer. The selection process and the reasons for full-text exclusion were recorded and documented in a PRISMA flow diagram [37].

### 2.4. Data Extarction and Analysis

The following items were extracted based on a previously designed Excel form: author, year, country, study design, participant’s characteristics, stimulation protocol, frequency and intensity, stimulation area, comparator, main outcomes, and follow-up. If relevant missing data was identified, the corresponding author was contacted and asked to provide the missing details. Data extraction was performed by one reviewer and checked by another. Any discrepancies were resolved through discussion. Given the high heterogeneity of the methods, protocols and stimulation area within the studies, meta-analysis was not carried-out and, thus, the results were merged and described narratively.

### 2.5. Quality Assessment

The risk of bias of the included studies were assessed with the Cochrane risk-of-bias tool for (RoB 2) [38] for RCTs, the Joanna Briggs Institute (JBI) checklist [39] for nRCT, and the Institute of Health Economics (IHE) quality appraisal checklist for case-series [40]. Quality assessment was performed independently by two reviewers and disagreements were solved by discussion or after consulting a third reviewer.

## 3. Results

A total of 353 studies were identified in the electronic databases. After removing duplicates, titles, and abstracts, 200 references were screened and 29 full-text articles were assessed for eligibility. Finally, 12 studies were included in this SR [13,15,17,20,41,42,43,44,45,46,47,48]. A list of ongoing studies can be found in Appendix A. Figure 1 shows the PRISMA flowchart of the study selection process. 

### 3.1. Study Characteristics

Table 2 shows the selected studies’ characteristics. Five of the included studies were RCT [13,15,41,42,44], one was an nRCT [47] and six were case-series [17,20,43,45,46,48]. None of the studies identified by this SR reported data on cost-effectiveness. Two different therapeutic protocols were identified within the studies: conventional high frequency (10–15 Hz) [13,17,20,41,43,44,45,46] and continuous or intermittent TBS [15,41,42,44]. Sample sizes ranged from 11–147 (median = 22.5). Stimulation areas were the prefrontal cortex (PFC) [47], bilateral PFC [41], medial PFC (MPFC) [15,44,46], left dorsolateral prefrontal cortex (DLPFC) [13,17,43,44,45,46,48] and cingulate cortex [44]. Intervention times fluctuated in a range of one to four weeks and follow-ups varied from one hour to eight months. 

### 3.2. Quality Assessment

#### 3.2.1. RCTs

The overall risk of bias was assessed with some concerns, except in one study [44], which enlisted overall high risk of bias. This is mainly because this study did not provide enough details on randomization, blinding process and selection of the reported results. Potential bias due to deviations from the intended intervention and selection of the reported results were the main source of bias. Figure 2 shows the risk of bias summary, with judgements about each risk of bias item for each included study. Figure 3 shows a graph with review authors’ judgements about each item presented as percentages across all included studies.

#### 3.2.2. nRCTs

The methodological quality of this study [47] was rated as high, since it met eight of nine criteria according to the JBI checklist. Only the one referring to multiple measurements before and after the intervention was negatively rated. The complete quality assessment of the included nRCT can be found in Appendix A.

#### 3.2.3. Case-Series

The methodological quality was rated as low in one study [46], high in another one [45] and moderate in the remaining four [17,20,43,48]. The study rated with lower quality was reported in a letter to the editor and therefore some information was incomplete or uncertain. Overall, the domains with the lowest scores were obtained in aspects related to the collection of cases in more than one centre, the recruitment of consecutive patients, the same level of disease within the participants, and the adequate reporting of losses during follow-up, which the studies did not provide information on. The complete quality assessment of the included case-series can be seen in Appendix A.

### 3.3. Effectiveness of TMS

Table 3 shows the results on effectiveness of the included studies. 

#### 3.3.1. Dorsolateral Prefrontal Cortex Stimulation

*rTMS over the left-DLPFC*, one RCT [13] and 4 case-series [20,43,45,46] applied high frequency stimulation (15 Hz), with intensity set at 100% of the resting motor threshold (RMT) and 2400 pulses per session, although with different numbers of sessions and timing. Another case-series [48] used an iTBS stimulation protocol, with 600 pulses per session and intensity set at 100%.

In Terraneo et al. (2016) [13] (*n* = 32) participants were randomized to rTMS (8 sessions over 4 weeks) or pharmacological treatment. At the end of the treatment period, there were more patients without relapses (no positive urine analysis) in the intervention group: 11 (69%) vs. 3 (19%) (OR = 6.47, IC95%: 1.14–36.6). Craving was also significantly lower with rTMS (*p* = 0.038), while no significant differences were found in depression. 

Madeo et al. (2020) [43] and Gómez et al. (2020) [20], from the same research group as Terraneo et al. [13], in two different retrospective studies, used an intensified version of the protocol used by the latter (10 sessions in the first five days, and subsequently two weekly sessions for 11 and 12 weeks, respectively). Cocaine consumption was assessed by a combination of urine analysis, self- and hetero report. Gómez et al. (2020) [20] (*n* = 87) observed a significant reduction in the number of days of consumption at the end of the treatment (from 19.2 to 0.8, t = 12.7, *p* < 0.001). This difference appeared from day 30. The same pattern of reduction was found in craving (Cocaine Craving Questionnaire, CCQ), sleep quality (Pittsburgh Sleep Quality Inventory, PSQI), depression (Beck Depression Inventory, BDI-II), anxiety (Self-rating Anxiety Scale, SAS) and global psychopathology (Symptoms Checklist-90-Revised, SCL-90-R) (all *p*-values < 0.001). Madeo et al. [43] (*n* = 284) retrospectively analysed patients followed for a range of 4 to 989 days. After receiving the intensified protocol, patients were treated with rTMS based on relapses or craving increase. The median time to relapse was 91 days (IC95%: 70–109), compared to 51 days (IC95%: 39–78) observed in a historical control group treated as usual (*n* = 173) (no statistical contrasts were performed). 

Pettorruso et al. (2019) [45] (*n* = 20) applied 24 sessions, 20 during the first two weeks, and two weekly sessions for the remaining two weeks. Out of the 16 participants who completed the treatment, 9 (56.2%) showed negative urine analysis (*p* = 0.003). Significant reductions were also obtained for craving (subscale of the Cocaine Selective Severity Assessment, CSSA), withdrawal symptoms (total score of the CSSA,), anxiety (SAS), depression (BDI-II), and global psychopathology (SCL-90- R) (all *p*-values < 0.02). Results did not reach the significance level for insomnia severity (*p* = 0.077, Insomnia Severity Index, ISI).

Politi et al. (2008) [46] (*n* = 36) applied 10 sessions, obtaining a significant reduction in craving for consumption (F = 4.96; *p* < 0.001).

Finally, Steele et al. (2019) [48] (*n* = 19) used an iTBS stimulation protocol. Treatment took place in three sessions per day, with approximately a 60-min interval between sessions, for 10 days over a two-week period (30 total iTBS sessions). Only 9 participants finished treatment and were followed for four weeks. According to self-reported measures, the weekly amount of money spent on cocaine and the number of days of consumption were reduced by 78% (*p* < 0.001) and 70% (*p* < 0.001), respectively. Craving was reduced by 37% when measured by the Cocaine Craving Questionnaire, and 26% when measured by the Cocaine Craving Scale. One week after treatment, scores in depression (Montgomery–Asberg Depression Rating Scale) were reduced by 18%, but a 33% increase was found in anxiety (Beck Anxiety Inventory), although scores were low in both cases.

High frequency deep rTMS over the bilateral PFC. A sham-controlled trial [41] (*n* = 18) and a case series [17] (*n* = 7) applied deep rTMS using a H1-coil. The stimulation was applied bilaterally, although with a preference for the left hemisphere. The intensity was set at 100%, and frequency was 10 Hz [41] and 20 Hz [17], respectively. In both studies, three weekly sessions were applied on alternate days for four weeks.

In Bolloni et al. (2016) [41], the effect on consumption reduction (evaluated by hair analysis) up to 6 months after treatment started was not significant (F = 0.35; *p* = 0.87). When groups were analysed separately, only the intervention group showed a significant reduction compared to baseline. Rapinesi et al. (2016) [17] observed a significant reduction in craving at the end of the treatment period (VAS 0–10, *p* < 0.001) and also four weeks later (*p* = 0.003).

#### 3.3.2. Medial Prefrontal Cortex (MPFC) Stimulation

Martínez et al. (2018) [44] (*n* = 18) randomized participants into three groups: high frequency deep stimulation (H7-coil), low frequency or sham stimulation. High frequency stimulation was delivered at 10 Hz and 1200 pulses per session. Low frequency was delivered using a standard 1 Hz protocol including 900 pulses per session. For the sham condition, a sham coil was present in the same TMS helmet. The intensity was progressively increased in both stimulation groups from 90% to 110% of the individual RMT. A choice test between administering cocaine or receiving money was performed at baseline, after four sessions and at the end of the treatment period. A significant interaction of treatment by occasion (F = 5.36, *p* = 0.02) was observed. There was little change in cocaine self-administration in the sham group or in the low frequency group across the three sessions. Only the high frequency group showed a decrease in the choice for cocaine, and this effect was manifested from the third session. However, craving, evaluated with a visual analogue scale, was not affected by any rTMS condition.

Two sham-controlled trials by Hanlon et al. (2015, 2017) [15,42] used a cTBS protocol (*n* = 11 and *n* = 25, respectively). A single session of cTBS and another session of sham stimulation were cross-applied, separated by 7–14 days. A total of 3600 pulses were applied over the MPFC, with intensity set at 110% of the individual RMT. Craving was assessed immediately after each session (VAS 0–10), while participants were exposed to cocaine-related cues. No significant differences were observed in both studies. In Hanlon et al. (2015) [15], when the change was analysed categorically (i.e., increase, decrease or no change), significantly fewer participants in the real stimulation session increased craving (χ^2^ = 5.64; *p* = 0.05).

#### 3.3.3. Comparison between Bilateral Deep Stimulation Protocols: iTBS vs. High Frequency rTMS

Sanna et al. (2019) [47], in a non-randomized trial (*n* = 47), compared bilateral deep stimulation (H4 coil) over the PFC and insula with two different protocols. One group received iTBS (600 pulses/session, 80% of the individual RMT) while the other group was treated with high-frequency rTMS (15 Hz, 2400 pulses/session, 100% of the individual RMT). In both groups, 20 sessions were applied for four weeks, with decreasing frequency. The results showed that the two stimulation protocols significantly reduced consumption measured by urine analysis (*p* <0.001 for the effect of time). No significant differences were found between protocols. At the end of the treatment period, 82% and 80% of patients on iTBS and rTMS, respectively, tested negative in the urine analysis. Craving (Weiss modified Cocaine Craving Questionnaire) and the risk of developing problems due to the use of cocaine (Alcohol, Smoking and Substance Involvement Screening Test, ASSIST) showed the same pattern of results. 

### 3.4. Evidence on the Safety of TMS for the Treatment of Cocaine Addiction

Table 4 describes dropouts and adverse events observed among the included studies. 

Drop-outs were relatively frequent in around 20% of the studies, although none specified that they were due to safety problems. Adverse events were not serious in any case.

Two studies provided detailed data on adverse events. In Madeo et al. (2020) [43], one case of seizure occurred in a 27-year-old woman 66 days after the first rTMS session. Another case of a hypomanic episode was reported in a 37-year-old man, just under 90 days after his first rTMS session. Twenty-three patients (8%) reported headache after stimulation, while the rest of the events, mild and transient, occurred in one or two patients each. In Steele et al. (2019) [48], one participant suffered right-hand supination/pronation at the wrist 10–15 min after the iTBS session. Two weeks after the iTBS termination, this same participant reported visual illusions and tactile hallucinations, which developed slowly over several days but were cleared promptly with a single dose of olanzapine. Nine of the 14 participants (64,3%) experienced at least one headache, usually beginning during or shortly after iTBS. 

## 4. Discussion

CUD is a disease that can cause cognitive dysfunctions at various levels, such as lack of impulse control, drug-seeking compulsions, and inability to modulate behaviours according to the different circumstances [45]. Today it is known that addictions to substances such as cocaine can compromise the activity patterns of the entire brain, and that their effects are focused on meso-cortical alterations and in the activity of dopamine, which affect the centres of motivation and desire to consume the substance [18].

In recent years, non-invasive brain stimulation techniques have provided insights into the neural networks affected by CUD and have been tested as an alternative to addiction treatments [17]. Such is the case of rTMS for which there is currently evidence that indicates a potential benefit in reducing the consumption of alcohol and other drugs [22]. This review makes a pioneering effort to collect evidence on the safety, effectiveness, and cost effectiveness of using different rTMS protocols on the reduction of cocaine consumption and reduction of craving.

What could be the mechanism by which rTMS produces beneficial effects on cocaine consumption? The honest answer is that we do not know, and, in any case, it seems to be out of the scope of a review article such as this. However, taking into account data obtained by mean of optogenetic stimulation in a rat model of cocaine addiction [12], it is tempting to speculate on the possibility that the activation of a hypofunctional prefrontal cortex (whose deep-layer pyramidal neurons project to subcortical structures implicated in drug-seeking behaviours, including the nucleus accumbens and dorsal striatum), produces a regulation on dysfunctional reward circuits. The results by Chen et al. clearly demonstrate two crucial aspects related to addiction. First, that consumption produces a reduction of activity in the cortex in cocaine-seeking rats and, second, that by increasing the excitability of the cortex, compulsive behaviour decreases. How rat prelimbic cortex stimulation reduces cocaine seeking remains to be resolved. An interesting possibility would be that the activation of the descending glutamatergic connections from the cortex might regulate the dopaminergic activity of accumbens and dorsal striatum. Such dopaminergic activity derives from the inputs arising in the ventral tegmental area and substantia nigra, and in cocaine addiction would have an abnormal dynamic to be corrected. In fact, it has been shown in humans that high-frequency rTMS of the left DLPFC induces dopamine release in the striatum [49]. Furthermore, high-frequency stimulation (rTMS at 20 or 25 Hz) delivered on the frontal cortex of rats induces dopamine release throughout the mesolimbic and mesostriatal circuits [50,51,52]. This suggests that the therapeutic benefit observed in humans using high-frequency TMS over the DLPFC could be related to a regulation of dopamine activity.

Moreover, it has been shown that 12-Hz optogenetic stimulation of medial prefrontal cortex projections to the nucleus accumbens dropped sensitivity to a cocaine challenge in mice. This specific protocol activates metabotropic glutamate receptors, which depotentiates excitatory inputs on dopamine D1 receptors [53].

Regarding safety, no serious adverse effects have been observed [43], and the most common adverse effect was mild and transient headache. These results are supported by the previous literature on rTMS in other clinical conditions (e.g., depression, OCD, other addictions) [22,54].

Although the included studies have reported favourable results in measures of consumption reduction and craving, the heterogeneity in central aspects, such as the evaluation criteria or the follow-ups, make it difficult to summarize and compare the results they offer. Additionally, only five of the 12 studies were RCTs, the rest were a controlled trial and six uncontrolled studies (with a significant percentage of losses in several of them) except in the case of two retrospective studies.

Despite this context of low-quality evidence, available results suggest that high-frequency rTMS applied to the left DLPFC may produce clinically relevant benefits in reducing cocaine use and craving, and possibly in other variables such as depression and insomnia [13,20,45,47,55]. As it was mentioned previously, this can be explained because the DLPFC participates in the reward, motivation and decision-making systems that allow inhibitory control [13,19]. However, the malfunction of the DLPFC (which may be abnormal in substance use disorders) may explain a reduction in inhibitory control and a greater probability of relapse into alcohol and drug use. Hence, high-frequency rTMS protocols with excitatory effect, located in the DLPFC, can provide better inhibitory control responses, as has been observed in the results of the studies included in this review.

These results are consistent with those reported by other studies in which a single session of repetitive TMS (rTMS) significantly reduced craving for cocaine, a reduction that persisted four hours after the end of the session [56]. Similarly, previous studies with stimulation protocols on bilateral DLPFC in other types of addictions have reported a reduction in nicotine consumption [55] and the desire for alcohol after cycles of deep transcranial magnetic stimulation (dTMS [17,57,58]). The evidence on the effects of other protocols (deep stimulation, TBS) is too limited to draw any conclusions, as is the evidence for stimulation of the medial PFC or other brain locations.

Although the use of TMS involves some difficulties, such as the need to transfer patients to centers that have such equipment, as well as the need to have physicians and experts in TMS, and protocols and technicians trained in handling the equipment, it is an option to consider in a scenario where there is a lack of approved treatment, and the fact that 70% of cocaine users seeking treatment relapse within the first three months, that cocaine dependent people have limited support to overcome this chronic illness [59].

### Limitations

Since not all studies are RCTs, and the fact that the included studies had small sample sizes and a short follow-up, the evidence obtained does not allow conclusive statements.

## 5. Conclusions

In this review, despite the low quality of the studies, significant first results of the efficacy of transcranial magnetic stimulation (TMS) have been observed in terms of the number of doses consumed and the reduction in craving to consume, with respect to the baseline values reported by the participants and by different biological tests (such as urine or hair). The ability to modulate cravings for use in a specific way through non-invasive brain stimulation techniques, such as rTMS, could be a new tool to use as an adjunct to the behavioural treatment of addiction, especially for cocaine use, in that there is currently no specific pharmacotherapy approved for its treatment. That said, in order to consider TMS as a procedure likely to be recommended for the treatment of cocaine addiction, controlled clinical trials are needed carried out under rigorous standards with respect, for example, to the characterization of the participants, the randomization and the blinding procedures. In this regard, there are a number of variables specific to the technique that make it difficult to obtain the “perfect” protocol for each subject and to obtain the best possible results. In the future, it will be necessary to conduct comparative studies to evaluate these key variables, among which it is necessary to mention the target region to be stimulated, the methods to locate the target and the type of coil to be used, the number and frequency of the pulses and the number of sessions. 

## Figures and Tables

**Figure 1 jcm-10-05595-f001:**
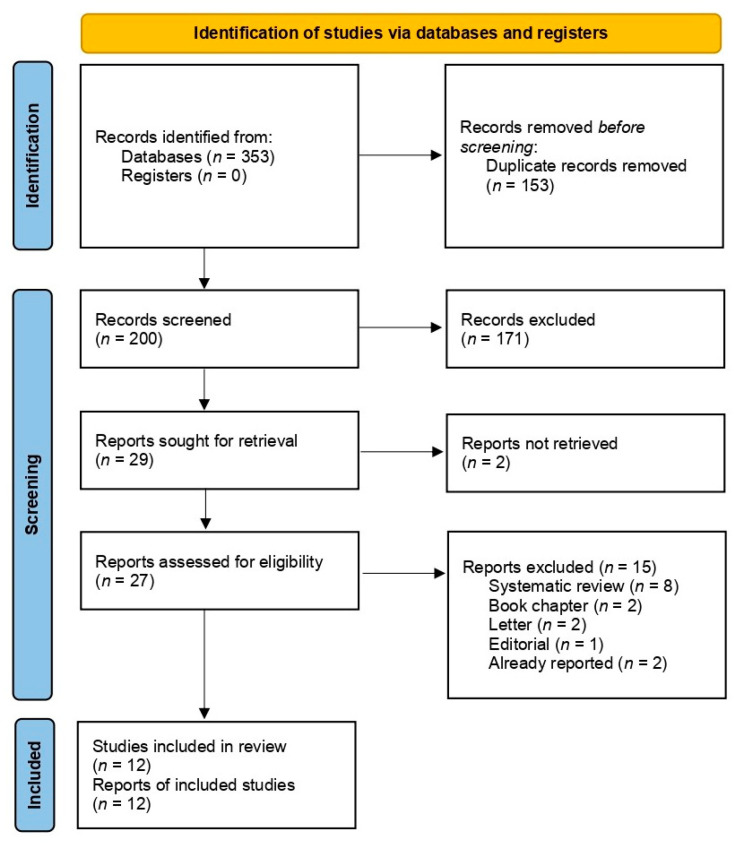
PRISMA flowchart of the study selection process.

**Figure 2 jcm-10-05595-f002:**
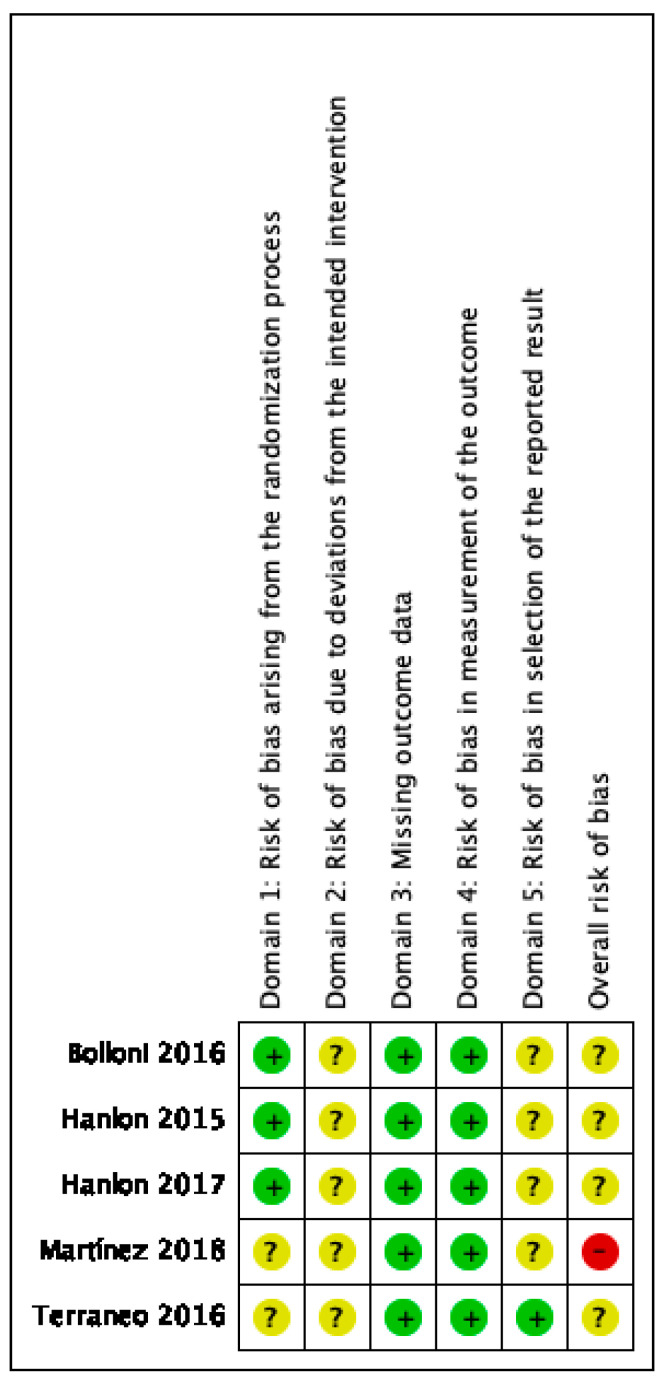
Risk of bias summary.

**Figure 3 jcm-10-05595-f003:**
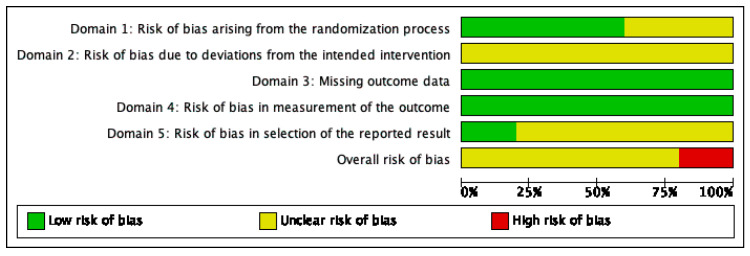
Authors’ judgements about each item.

**Table 1 jcm-10-05595-t001:** Medline search strategy.

1	Transcranial Magnetic Stimulation	11,631
2	(Transcranial adj1 magnetic stimulation$).tw.	14,512
3	((transcranial magnetic stimulation or tms) adj5 repetitive).tw.	4917
4	((transcranial magnetic stimulation or tms) adj5 rhythmic).tw.	41
5	(rtms or tms).tw.	15,423
6	((Repetitive or “single pulse” or “paired pulse”) adj1 “transcranial magnetic stimulation”).tw.	5260
7	1 or 2 or 3 or 4 or 5 or 6	21,895
8	Cocaine-Related Disorders	8278
9	(cocaine * adj2 (abuse* or addict * or dependent * or disorder *)). ti,ab.	7622
10	8 or 9	12,519
11	7 and 10	46

**Table 2 jcm-10-05595-t002:** Characteristics of the included studies.

Author (Year), Country	Study Design	Population	No. of Participants	Intervention	Stimulation Area	Stimulation Protocol	Frequency (Hz) and Intensity (% RMT)	Comparator	Outcomes	Measures
Bolloni (2016) [41], Italy, USA and Israel	RCT	CUD (DSM-5) Male: *n*=16 Female: *n* = 2 Mean age: 27–48 years	*n* = 18	Deep rTMS	Bilateral PFC	12 rTMS sessions were administered three times a week for 4 weeks	10 Hz 100–120%	Sham	Cocaine intake (hair analysis)	Baseline, after 1 month and 3 and 6 months later
Gómez (2020) [20], Italy	CS	CUD (DSM-5) Male: *n* = 85 Female: *n* = 2 Mean age: 37.67 years	*n* = 87	rTMS	Left-DLPFC	2 sessions per day for the first 5 consecutive days of treatment (10 sessions), and 2 sessions per week for the following 12 weeks.	15 Hz 100%	NA	Cocaine use (self-report and urine screens) Craving (CCQ) Sleep Quality (PSQI) Depression (BDI-II) Anxiety (SAS) Symptoms (SCL-90)	Baseline, and after 5, 30, 60, and 90 days of rTMS treatment.
Hanlon (2015) [15], USA	RCT	Cocaine users Male: *n* = 9 Female: *n* = 2 Mean age: 39 years	*n* = 11	cTBS	MPFC	2 stimulation visits (occurring within 7–14 days of each other).	5 Hz 110%	Sham	Craving (VAS)	Before and after the cTBS session
Hanlon (2017) [42], USA	RCT	Cocaine users Male: *n* = 12 Female: *n* = 3 Mean age: 42 years	*n* = 25	cTBS	MPFC	2 stimulation visits (occurring within 7–14 days of each other) with exposure to 6 trains of cBTS.	5 Hz 110%	Sham	Craving (VAS)	Before and after the cTBS session
Madeo (2020) [43], Italy and USA	CS	CUD (DSM-5) Male: *n* = 139 Female: *n* = 8 Mean age: 36.6 years	*n* = 147	rTMS	Left-DLPFC	2 rTMS sessions per day for the first 5 days, then weekly, twice per day on each session day for 11 consecutive weeks.	15 Hz 100%	NA	Cocaine use (urine screening, self-report and reports by collateral informants)	Baseline, after 3 months of rTMS and up to 2 years.
Martínez (2018) [44], USA and Israel	RCT	CUD (DSM-5) Male: *n* = 17 Female: *n* = 1 Mean age: 43.3 years	*n* = 18	rTMS	MPFC	The rTMS was delivered on weekdays, over the course of 3 weeks.	HF: 10 Hz LF: 1 Hz 90–110%	Sham	Number of doses chosen during self-administration Craving (VAS)	Baseline, after 4 days and after 13 days of rTMS.
Pettorruso (2019) [45], Italy and UK	CS	CUD (DSM-5) Male: *n* = 4 Female: *n* = 2 Mean age: 36.63 years	*n* = 16	rTMS	Left-DLPFC	20 stimulation sessions (2 daily, 5 d/week) for 2 weeks and 2 consecutive maintenance rTMS sessions once a week for 2 weeks.	15 Hz 100%	NA	Cocaine use (urine test) Cocaine withdrawal signs and symptoms (CSSA) Craving (CSSA) Depression (BDI) Anxiety (SAS) Global Psychopathology (SCL-90) Insomnia (ISI)	Baseline, after 2 and after 4 weeks of rTMS treatment.
Politi (2008) [46], Italy	CS	CUD (DSM-IV) Male: *n* = 31 Female: *n* = 5 Mean age: NI	*n* = 36	rTMS	Left-DLPFC	10 daily sessions of rTMS.	15 Hz 100%	NA	Craving (VAS)	During sessions of rTMS.
Rapinesi (2016) [17] Italy	CS	CUD (DSM-IV) Male: *n* = 7 Female: *n* = 0 Mean age: 48.71 years	*n* = 7	Deep TMS	Bilateral PFC	3 weekly sessions on alternate days for 4 consecutive weeks, for a total of 12 sessions.	20 Hz 100%	NA	Craving (VAS)	Baseline and after 2, 4 and 8 weeks of treatment.
Sanna (2019) [47], Italy	nRCT	CUD (DSM-5) Male: *n* = 45 Female: *n* = 2 Mean age: 37.40 years	*n* = 47	iTBS	PFC	20 stimulations over 4 weeks: 10 stimulations during the 1st week, 4 stimulations during the 2nd week, 3 stimulations during the 3rd and 4th week.	HF rTMS: 15 Hz; 100% iTBS: 5 Hz; 80%	HF rTMS	Cocaine use (urine test) Craving (CCQ-brief) Risk for developing problems due to the use of cocaine (ASSIST)	Baseline, weekly during treatment and at the end of treatment.
Steele (2019) [48], USA	CS	CUD (DSM-5) Male: *n* = 13 Female: *n* = 6 Mean age: 47.4 years	*n* = 19	iTBS	Left-DLPFC	3 iTBS sessions per day, with an interval of approximately 60-min between sessions, for 10 days over a 2-week period (30 total iTBS sessions).	5 Hz 90–120%	NA	Amount of money spent on cocaine consumption Craving (CCS and CCQ) iTBS side effects Depression (MADRS)	Baseline, during, and after the intervention and at 1-and 4-week follow-up visits.
Terraneo (2016) [13], Italy	RCT	CUD Male: *n* = 30 Female: *n* = 2 Mean age: 40.28 years	*n* = 32	rTMS	Left-DLPFC	1 rTMS session per day during the first 5 days of treatment, and then once a week for the following 3 weeks, for a total of 8 rTMS sessions.	15 Hz 100%	Pharmacological agents	Cocaine use (urine test) Craving (VAS) Adverse events Depression (SCL-90)	Baseline, after 29-day treatment and after 63-day follow-up.

ASSIST: Alcohol, Smoking and Substance Involvement Screening Test; BDI-II: Beck Depression Inventory-II; cBTS: continuous theta burst stimulation; CCQ: Cocaine Craving Questionnaire; CCQ: Cocaine Craving Questionnaire; CCSA: Cocaine Selective Severity Assessment; CIP: Cocaine-Induced Psychosis Screener; CS: case-series; CUD: cocaine use disorder; DLPFC: dorsolateral prefrontal cortex; DSM: Diagnostic and Statistical Manual of Mental Disorders; HF: high frequency; ISI: Insomnia Severity Index; cTBS: continuous theta burst stimulation; iTBS: intermittent theta burst stimulation; LF: low frequency; MADRS: Montgomery–Asberg Depression Rating Scale; MPFC: medial prefrontal cortex; NA: Not applicable; NA: not applicable; RCT: randomized controlled trial; NI: No Information; nRCT: non-randomized controlled trial; PFC: prefrontal cortex; PSQI: Pittsburgh Sleep Quality Index; RMT: resting motor threshold. rTMS: repetitive transcranial magnetic stimulation; SAS: Self-rating Anxiety Scale; SAS: Zung Self-Rating Anxiety Scale; SCL-90: Symptom checklist 90-revised; UK: United Kingdom; USA: United States of America; VAS: visual analogue scale.

**Table 3 jcm-10-05595-t003:** Effectiveness results among the included studies.

	Cocaine Use	Craving	Anxiety	Depression	Psychopathology	Insomnia
rTMS over the left-DLPFC						
Terraneo et al., 2016 [13] (*n* = 36) RCT	Negative urine test during treatment: rTMS: 11(69%) PT: 3 (19%) OR = 6.47 (IC95%: 1.14, 36.6).	VAS 0–10: Significantly lower craving with rTMS ANOVA RM: F (1,27) = 4.74, *p* = 0.038	-	-	-	-
Madeo et al., 2020 [43] (*n* = 147) CS	Days until relapse (median): rTMS: 91 (70–109) TAU: 51 (39–78) 147 patients followed for 84–974 days: Mean use <1.0 day/month (median 0).	-	-	-	-	-
Gómez et al., 2020 [20] (*n* = 87) CS	Days of cocaine use (mean): reduction at 30 days: −18.7 (97.3%) *p* < 0.001 reduction at 90 days: −18.3 (95.6%) *p* < 0.001	CCQ reduction at 30 days: −11.32 (89.3%) *p* < 0.001 reduction at 90 days: −8.86 (69.9%) *p* < 0.001	SAS reduction at 30 days: −11.96 (24.9%) *p* < 0.001 reduction at 90 days: −9.83 (20.5%) *p* < 0.001	BDI-II reduction at 30 days: −13.89 (73.1%) *p* < 0.001 reduction at 90 days: −12.26 (64.5%) *p* < 0.001	SCL-90-R reduction at 30 days: −18.24 (27.7%) *p* < 0.001 reduction at 90 days: −19.45 (29.5%) *p* < 0.001	PSQI reduction at 30 days: −4.24 (45.6%) *p* < 0.001 reduction at 90 days: −3.12 (33.8%)
Pettorruso et al., 2019 [47] (*n* = 20) CS	Negative urine test at the end of treatment: 9 of 16 (56.25%) (Z = −3.00; *p* = 0.003).	CSSA (craving) reduction at 4 weeks: −1.5 (33.9%) *p* = 0.020	SAS reduction at 4 weeks: −8.4 (23.0%) *p* = 0.001	BDI-II reduction at 4 weeks: −9.8 (57.1%) *p* = 0.008	SCL-90-R reduction at 4 weeks: −0.51 (52.0%) *p* < 0.001	ISI reduction at 4 weeks: −5.2 (59.7%) *p* = 0.077
Politi et al., 2008 [48] (*n* = 36) CS	-	Greater reduction with TMS ANOVA RM (time effect) F (30,270) = 4.96 *p* < 0.001	-	-	-	-
Steele et al., 2019 [42] (*n* = 19) CS	Mean use (days/week): Reduction at 7 weeks: −3 (70.0%) *p* < 0.001 Money spent on consumption at 4 weeks: −167$ (78.0%) *p* < 0.001	Reduction at 7 weeks: CCQ: 37% CCS: 26%	BAI Increase at 3 weeks: 33%	MADRS Reduction at 3 weeks: 18%	-	-
**High frequency deep rTMS over the bilateral PFC**						
Bolloni et al., 2016 [41] (*n* = 18) RCT	Amount of cocaine in hair analysis ANOVA RM (interaction) F = 0.35; *p* = 0.87	-	-	-	-	-
Rapinesi et al., 2016 [17] (*n* = 20) CS	-	VAS 0–10 Craving reduction at the end of treatment: −6.3 (64.7%) *p* < 0.001 Craving reduction at one month follow-up: −3.8 (39.6%) *p* = 0.003	-	-	-	-
**Medial prefrontal cortex (MPFC) stimulation**						
Martínez et al., 2018 [46] (*n* = 18) RCT	Choice of cocaine vs. receiving money: lower with rTMS ANOVA RM (interaction) F = 5.36, *p* = 0.02	Negative binominal distribution with random effects F (2, 14) = 0.77, *p* = 0.48	-	-	-	-
Hanlon et al., 2015 [15] (*n* = 11) Crossover	-	VAS 0–10 No significant different on mean change. Fewer patients getting worse and more patients remaining stables with TMS. χ2 = 19.14, *p* <0.001	-	-	-	-
Hanlon et al., 2017 [44] (*n* = 25) Crossover	-	VAS 0–10 Post-treatment result not significant (*p*-value not reported): rTMS: 2.93 (2.78) Control: 2.90 (2.25)	-	-	-	-
**Comparison between bilateral deep stimulation protocols: iTBS vs. high frequency rTMS**						
Sanna et al., 2019 [41] (*n* = 49) nRCT	Urine test and consumption statement ANOVA RM: Significant effect of time (F = 49.97; *p* <0.001) but not of treatment (F = 0.67) or interaction (F = 0.66).	brief modified CCQ ANOVA RM: Significant effect of time (F = 127.3; *p* <0.001 but not of treatment (F = 1.48) or interaction (F = 0.03).	-	-	-	-

BAI: Beck Anxiety Inventory; BDI-II: Beck Depression Inventory-II; CCQ: Cocaine Craving Questionnaire; CS: case series; CSSA: Cocaine Selective Severity Assessment; DLPFC: dorsolateral prefrontal cortex; ISI: Insomnia Severity Index; MADRS: Montgomery–Asberg Depression rating scale; nRCT: non-randomized controlled trial; OR: odds ratio; PSQI: Pittsburgh Sleep Quality Index; PT: pharmacological treatment; RCT: randomized controlled trial; RM: repeated measures; rTMS: repetitive transcranial magnetic stimulation; SAS: Self-rating Anxiety Scale; SCL-90-R: Symptoms Checklist 90 revised; TAU: treatment as usual; VAS: visual analogue scale.

**Table 4 jcm-10-05595-t004:** Dropouts and adverse events.

Author (Year)	Drop-Outs	Adverse Events
Boloni (2016) [41]	4/18 (22.2%)	Discomfort was not observed except for a patient who suffered from a mild headache after receiving active stimulation.
Gómez (2020) [20]	NR	Serious AEs were not reported. There were no seizures, syncopes, neurological complications or subjective complaints about memory or concentration impairment limiting the treatment.
Hanlon (2015) [15]	NA	NR
Hanlon (2017) [42]	NA	NR
Madeo (2020) [43]	58/284 (20.4%)	AEs were reported by 41 of the 284 patients. AEs reported were headache (*n* = 23), hypomania (*n* = 4), anxiety (*n* = 2), irritability (*n* = 2), dental pain (*n* = 2), scalp discomfort during the first 2 (*n* = 1), angioedema and urticaria (*n* = 1), distractibility (*n* = 1), dizziness (*n* = 1), nausea (*n* = 1), nausea and numbness (*n* = 1), seizure (*n* = 1), and a hypomanic episode (*n* = 1).
Martínez (2018) [44]	NR	Participants had difficulty tolerating stimulation that increased from 100 to 120% of MT, and thus the protocol was amended by lowering the maximal stimulation.
Pettorruso (2019) [45]	4/20 (20%)	The treated subjects reported no significant side effect.
Politi (2008) [46]	NR	NR
Rapinesi (2016) [17]	0/7 (0%)	All patients tolerated the stimulation without complications or AEs.
Sanna (2019) [47]	4/47 (8.5%)	A few participants in both the 15 Hz rTMS and the iTBS groups reported mild discomfort at the start of stimulation, especially during the first session. Both treatments were safe and there were no serious or unexpected AEs related to the treatments. There were no seizures or any other transient neurological event.
Steele (2019) [48]	7/16 (43.7%)	There were no unexpected, serious AEs. Nine of the 14 participants experienced at least one headache. One participant experienced sudden pain around her eyes and one experienced muscle soreness in the right forearm. No negative side-effects in cognitive and affective assessments were reported. No participant experienced any signs of mania or suicidality. After completing 26 iTBS sessions, a participant reported right-hand supination/pronation and thus treatment was terminated.
Terraneo (2016) [13]	3/32 (9.4%)	A few participants reported mild discomfort at the start of stimulation, especially during the first session, but overall, there were no significant differences in AEs across groups. There were no serious or unexpected AEs.
AEs: adverse events; NR: not reported; NA: not applicable.		

## Data Availability

The data presented in this study are available in Appendix A.

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
