# Peer review of "Transcranial Magnetic Stimulation for the Treatment of Cocaine Addiction: A Systematic Review"

_jcm, 2021, doi:10.3390/jcm10235595_

Round 1

Reviewer 1 Report

This is an interesting review. and I would make a few recommendations:

  1. Edit the review of TMS to more clearly describe differences in types of TMS, including rTMS vs. dTMS vs. TBS. 
  2. Given the role of dopamine in cocaine use disorder, it is worth discussing TMS effects on brain dopamine and implications for treatment.
  3. You cannot say TMS is a promising treatment for cocaine use disorder after saying the quality of the publications used in the review is poor.  I recommend more discussing the theoretical aspects of why TMS may be beneficial and contrast this with the state of research--this will provide information on gaps in knowledge and guide clinical treatment and future research.  

Reviewer 2 Report

This review provides a well-conducted overview of the available studies regarding rTMS use in cocaine use disorder. It is very well-written, well-structured, and offers a good insight on the available literature. I have only two concerns:

-A whole year has passed since the last search and the submission of the manuscript. I know it may be tedious but please do search for any studies published in the last year.

-It would be interesting to state whether there are any ongoing trials on this matter (e.g. clinicaltrials.gov), whose results should be expected.
